# Pyrrole-2-carboxaldehydes: Origins and Physiological Activities

**DOI:** 10.3390/molecules28062599

**Published:** 2023-03-13

**Authors:** Seiichi Matsugo, Yutaka Nakamura

**Affiliations:** 1Department of Applied Life Sciences, Faculty of Applied Life Science, Niigata University of Pharmacy and Applied Life Sciences, Higashijima 260-1, Akiha, Niigata 956-8603, Japan; 2Kanazawa University, Kakuma, Kanazawa 920-1192, Japan

**Keywords:** pyrrole-2-carboxaldehyde (Py-2-C), fungi, plant, natural product, physiological activity, structure–activity relationship

## Abstract

Pyrrole-2-carboxaldehyde (Py-2-C) derivatives have been isolated from many natural sources, including fungi, plants (roots, leaves, and seeds), and microorganisms. The well-known diabetes molecular marker, pyrraline, which is produced after sequential reactions in vivo, has a Py-2-C skeleton. Py-2-Cs can be chemically produced by the strong acid-catalyzed condensation of glucose and amino acid derivatives in vitro. These observations indicate the importance of the Py-2-C skeleton in vivo and suggest that molecules containing this skeleton have various biological functions. In this review, we have summarized Py-2-C derivatives based on their origins. We also discuss the structural characteristics, natural sources, and physiological activities of isolated compounds containing the Py-2-C group.

## 1. Introduction

The factors that contribute to a healthy lifestyle are a topic of interest in modern society. Some diseases are related to lifestyle, including diabetes, heart disease, and cancer. Preventive treatment of such diseases is important. Lifestyle diseases are sometimes strongly connected with a person’s diet, and smoking, binge eating, and drinking are strong risk factors for these diseases. Early detection of lifestyle diseases is important and many biomarkers have been developed in recent years for this purpose.

Many biomarkers have been reported for the early detection and determination of the stage of diabetes. In the stages of diabetes, many metabolites are produced by the reaction of glucose and amino acids. This process occurs via the well-known Amadori and Maillard reactions. After a complicated series of reactions, advanced glycation end products (AGEs) are produced. Typical AGEs include carboxymethyllysine, carboxyethyllysine, GA-pyridine, glucospan, arg-pyrimidine, pentosidine, and pyrraline. Pyrraline (**1**) has a structure that contains a formyl functional group at the 2-position of a pyrrole ring. Interestingly, AGEs are also produced in fermentation procedures. This finding suggests that similar reaction processes might take place under human physiological conditions over time, which is in agreement with the observation that lifestyle diseases mainly occur in the elderly. If such fermentation processes are also occurring over time in other biological systems, AGEs might be observed in other organisms. To investigate this hypothesis, we focused our attention on Py-2-C and examined naturally occurring Py-2-C derivatives. Furthermore, we discuss the biological activities of Py-2-C derivatives, including both in vitro and in vivo studies. We also explore the origins of Py-2-C by summarizing recently reported synthetic and biosynthetic routes for this compound.

## 2. Synthesis and Biosynthesis of Pyrrole-2-carbaldehydes

The chemical condensation of glucose and alkylamines under acidic conditions has produced *N*-substituted-5-methoxymethyl-2-formyl-pyrroles (**2**) in low yields (for **2b**, isolated yields < 5%) [1]. This synthesis was the first example of the preparation of a Py-2-C. The reaction was performed in 1 h under the catalysis of acetic acid, which suggested that this type of reaction might occur under physiological conditions over time (days, months, or years). However, many other reaction products were produced under the reaction conditions (as determined by thin-layer chromatographic analysis, but compounds were not isolated), and separation and purification of the hydroxymethyl derivatives from other byproducts were quite difficult. Under the reaction conditions, 2,4-dinitrophenylhydrazine (DNP) was used to stabilize the reaction products; however, the use of DNP caused further reactions, such as dehydration. This reaction procedure is also applicable to l-lysine. The condensation reaction of l-lysine monohydrochloride with glucose at elevated temperature (105 °C) for 6 h afforded ε-(2-formyl-5-hydroxymethyl-pyrrol-1-yl)-l-norleucine (**1**) at a 0.16% yield based on l-lysine [2]. Py-2-Cs were not isolated under these reaction conditions; however, this type of reaction might also take place in vivo. Under similar reaction conditions, several amino acids have been reacted with glucose (Maillard reaction), and the reducing ability of the products was examined. In the reaction of a γ-amino acid and glucose, γ-(2-formyl-5-hydroxymethyl-pyrrol-1-yl)-butyric acid (**3**) was isolated in low yield [3]. Among the products of the Maillard reaction of 10 amino acids and d-glucose, GABA showed the highest ferricyanide reduction potential (ascorbic acid was used as a standard). However, precise examination of the reducing ability of these Maillard reaction products revealed that the reducing ability reached the highest reducing state at 0.5 h after the start of the Maillard reaction. At this time, the formation of γ-(2-formyl-5-hydroxymethyl-pyrrol-1-yl)-butyric acid did not correspond with the reducing ability. The formation of γ-(2-formyl-5-hydroxymethyl-pyrrol-1-yl)-butyric acid gradually increased after 1 h and reached a constant level 10 h after the start of the Maillard reaction. This reaction profile suggested the possibility that other Maillard reaction products were produced in the early stages of the reaction, which were responsible for the reducing ability in the early stages. While, the formation of γ-(2-formyl-5-hydroxymethyl-pyrrol-1-yl)-butyric acid might decrease the reducing ability. Bearing a formyl group side chain, γ-(2-formyl-5-hydroxymethyl-pyrrol-1-yl)-butyric acid has a potential reducing ability for metal ions; however, the reducing ability is not strong enough to reduce metal ions completely. The proposed mechanism for the formation of Py-2-C from glucose and alkylamines (including lysine and γ-butyric acid) has been suggested to occur via the formation of 3-deoxy-d-glucose in the reaction course. In this mechanism, the alkylamine attacks the enol site of 3-deoxy-d-glucose forming an enamino diketone. Then, intramolecular cyclization of the amine moiety with the ketone produces a dihydropyrrole bearing formyl and hydroxy groups on the ring. Dehydration of the dihydropyrrole produces a Py-2-C as the final product. In the synthetic reaction, the initial attack of the alkylamine on the formyl group of glucose (in the open form of glucose) is the trigger for initiating the reaction. The hydration of the imine moiety to a carbonyl compound provided 3-deoxy-d-glucoside as a key intermediate, which was smoothly dehydrated to afford the enone. The Michael addition of the alkylamine to this enone and subsequent ring closure provided the dihydropyrrole. The dehydration of the dihydropyrrole (aromatization) took place to afford Py-2-C (**4**) as the final product [4,5,6] (Figure 1). The total synthesis of pyrrolemarumine (**5**) was attained by the regioselective functionalization of 2-formylpyrrole [7]. The biosynthesis of pyrrole consists of amino acids (glycine, proline, serine, threonine, and tryptophan) and dicarboxylic acids (malonic acid, oxaloacetic acid, and succinic acid) as synthons for the construction of the aromatic ring [8]. Another biosynthetic pathway for the formation of a pyrrole ring has been recently reported by Lautru et al. [9]. In this biosynthetic pathway, *N*-acetylglucosamine was used as the precursor, and after several enzymatic processes, 4-acetamidepyrrole-2-carboxylic acid was isolated. Further reactions were performed to afford congocidine as the final product. Very recently, the biosynthesis of a Py-2-C was attained via enzymatic fixation [10] in which the production of Py-2-C (**6**) from pyrrole was achieved using *Pseudomonas aeruginosa* HudA/PA0254 and *Segniliparus rotundus* CAR (Figure 1). 

## 3. Naturally Occurring Pyrrole-2-carbaldehyde Derivatives

### 3.1. Pyrrole-2-carboxaldehydes from Fungi

Edible mushrooms are very popular in eastern countries such as China, Korea, and Japan. Edible mushrooms contain many nutrients, including amino acids, proteins, lipids, and carbohydrates. Some mushrooms have been reported to have pharmacological activities. Identification of the key compounds from the extracts of edible mushrooms has attracted much attention from pharmacological companies and scientists.

From the mangrove plant *Aegiceras corniculatum*, two Py-2-C derivatives (**7** and **8**) and one indole alkaloid were isolated, and the structures were determined with spectroscopic analyses [11]. Extraction of the freeze-dried powder of *Mycoleptodonoides aictchisonii* resulted in the isolation of a Py-2-C bearing a carboxylic acid side chain. This compound was isolated by a series of extraction procedures, including hot methanol extraction, filtration, and subsequent extraction with ethyl acetate. The ethyl acetate layer was separated via silica gel column chromatography followed by HPLC separation using an ODS column. The Py-2-C bearing α-*iso*-butyl-acetic acid at the *N*-position was isolated along with 10 other compounds. The chemical structure of this Py-2-C (**9**) was determined by spectroscopic analyses [12]. Pyrrolezanthine (**10**) was first isolated from the edible mushroom *Leccinum extremiorientale* in 2011 [13]. The structure of pyrrolezanthine (**10**) was confirmed by a comparison of the NMR and MS data with the literature data [14]. From the cultivation of *Annulohypoxylon lianense*, the pyrrole alkaloid ilanepyrrolal (**11**) was isolated and characterized based on the spectroscopic analyses [15]. From the cultivation of *Chaetomium globosum*, which is an endophytic fungus in *Ginkgo biloba*, *N*-unsubstituted-5-hydroxymethyl-Py-2-C (**4**) was isolated and characterized [16]. In 2014, from an extract of the edible mushroom makomotake (*Zizaria latifolia*), three Py-2-Cs have been isolated and identified. Two of the Py-2-Cs (**12** and **13**) had already been reported (one compound (**13**) had already been chemically synthesized [5]) [5,17]; however, the Py-2-C (**14**) was a new compound [18]. The structures of the two known compounds were determined by a comparison with the previously reported spectral data. The molecular formula of the new compound was determined by HRESIMS and the molecular structure was determined using ^1^H- and ^13^C-NMR spectra, including DEPT, COSY, HMQC, and HMBC techniques. From the endophytic fungus of *Xylaria papulis*, a Py-2-C (**15**, popupyrrolal) was isolated and characterized based on the spectroscopic analyses [19]. *Cordyceps* species have been widely used in traditional medicine, especially for their effects on metabolic pathways. From an extract of *Cordyceps militaris*, 12 known compounds (including pyrrole alkaloids and nucleotide derivatives) and two new compounds, named cordyrrole A and B, were isolated and characterized. Cordyrrole A (**16**) is a Py-2-C derivative bearing a six-membered cyclic amide at the *N*-position of the pyrrole ring [20]. In 2015, from the edible mushroom, *Xylaria nigripes,* four Py-2-Cs (**17**–**20**) were isolated and identified. Two pyrrole alkaloids had already been reported: the Py-2-Cs pollenpyrroside A (**17**) and acortatarin A (**18**). The other two Py-2-Cs (**19** and **20**) have a tricyclic ring structure comprising a common bicyclic system with 2-formyl-pyrrole and substituted morpholine rings and a ketohexoside ring [21]. From the fruiting bodies of *Leccinum extremiorientale*, two known Py-2-Cs bearing a butyric acid moiety (**3** and **21**) and one unknown Py-2-C bearing an acetic acid moiety (**22**) were isolated and characterized based on 1D- and 2D-NMR spectroscopy and MS [22]. All of these compounds are carboxylic acid derivatives connected by a short-chain (one- or three-carbon) linker to the *N*-position of the pyrrole ring. From the ethanol extract of the fermented mycelia of *Xylaria nigripes*, two unknown (**23** and **24**) and six known Py-2-Cs were isolated along with flavonoid derivatives. The chemical structures of the known Py-2-Cs (**25**, **26**, **27**, **3**, **21**, and **12**) were determined based on the spectroscopic analyses [23]. From the endophytic fungus *Mollisia* sp., among many metabolites, one known Py-2-C (**15**) was isolated and identified [24]. From the edible mushroom *Basidiomycetes-X* (*Echigoshirayukitake*), three Py-2-C derivatives (**3**, **4**, and **28**) have been isolated and identified [25]. Two compounds had already been reported and one was a new compound (**28**) bearing a carbamide functional group on the side chain. From another edible mushroom, *Phlebopus portentosus*, three unknown (**29**, **30**, and **31**) and four known Py-2-Cs (**32**–**35**) were isolated and the molecular formulas of each compound were determined with HRESIMS. The structure determination of the compounds was based on NMR spectroscopic analyses, including COSY and HMBC [26]. Among the known compounds, two compounds (**33** and **35**) had been previously isolated from the mycelium of *Inonotus obliquus* [27,28]. One compound (**34**) was a *Ganoderma* alkaloid previously isolated from the hypha of deep fermented *Ganoderma capense* [29]. Compound (**33**) had already been isolated from *Lyceum Chinense* in 2013 [30] (Figure 2).

Very recently, in a study on the metabolites of *Purpureocillium lavendulum* produced against *Meloidogyne incognita*, seven metabolites were isolated and structurally characterized. These compounds were five steroid derivatives, one alloxazine derivative, and 5-methoxymethyl-1*H*-Py-2-C (**36**) [27] (Figure 2).

### 3.2. Pyrrole-2-carbaldehydes from Plant Sources

Py-2-Cs have been shown to be some of the compounds responsible for the aromatic flavor of raw cane sugar using GC-MS [31]. The detected Py-2-Cs (**37**–**40**) all have a cyclic lactone structure, which is produced by the condensation reaction of the hydroxymethyl group at the 5-position of the pyrrole ring with the carboxylic acid of the amino acid. From roasted chicory root (*Cichorium intrybus* L.), three Py-2-C lactones (**38**–**40**) were isolated and the structures were determined based on the spectroscopic analyses (NMR, IR, UV, and MS) [32]. A butanol extract of flue-cured Virginia tobacco was fractionized to afford an acidic dichloromethane-soluble fraction. From this fraction, an unknown 2-formylpyrrole acetic acid derivative (**41**) was isolated and the structure was determined based on the spectroscopic analyses [33]. From the root of *Pisum saticum*, a Py-2-C (**42**) bearing a five-membered lactone at the *N*-position was isolated and identified based on the spectroscopic analyses [34]. This Py-2-C acted as a specific regulator of trigonelline-induced G2 arrest. Py-2-Cs have also been isolated from other sources. For example, from the odorous flower *Quararibea funebris*, the Py-2-C funebral (**43**) with a lactone chromophore was isolated and characterized by various spectroscopic methods [35]. The total synthesis of funebral was first reported in 1999, which used the Paal–Knorr condensation as a key step to prepare the pyrrole lactone moiety [36]. Another total synthesis of funebral has also been reported that used a chiral nitrone as a key substrate [37]. From the leaves of *Magnolia coco*, mangnolamide (**44**) with a phenylpropanoid chromophore has been isolated and characterized using HREIMS, IR, MS, and ^1^H- and ^13^C-NMR spectroscopy [38]. The exact structure was determined using NOE and HMBC spectra along with the MS fragmentation pattern. The total synthesis of magnolamide was achieved in six steps starting from the condensation of pyrrole-2,5-dicarboxaldehyde with *N*-(4-bromobutyl) phthalimide [36]. From the Formosan plant *Zanthoxylum simulans*, the pyrrole alkaloid pyrrolezanthine (**10**, 5-hydroxymethyl-1[2-(4-hydroxyphenyl)-ethyl]-1*H*-Py-2-C) was isolated along with the previously unknown (–)-simulanol, zanthopyranone, and 28 known compounds [14]. The structure of pyrrolezanthine (**10**) was determined based on the spectroscopic analyses. Among the unsaturated fatty acids isolated from the seed of *Allium fistulosum* L., 4-(2-formyl-5-hydroxymethylpyrrol-1-yl) butyric acid (**3**) was identified [39] and the structure was determined based on a previous report [3]. From the seeds of the sweet chestnut *Castanea sativa*, methyl-(5-formyl-1*H*-pyrrole-2-yl)-4-hydroxybutyrate (**45**) was isolated and the structure was identified based on the spectroscopic analyses [40]. Two new pyrrole alkaloids were isolated from *Bolbostemma paniculatum* [41] and identified as 4-(2-formyl-5-methoxymethylpyrrol-1-yl) butyric acid (**3**) and methyl 2-(2-formyl-5-methoxymethylpyrrol-1-yl)-3-phenylpropionic acid (**46**). From the ethanol extract of prunes (*Prunus domestica* L.), the bipyrrole carboxaldehyde (**47**) was isolated and identified by spectroscopic analysis (^1^H-, ^13^C-NMR, HMBC, IR, UV, and HR-FABMS) to be 2-(5-hydroxymethyl-2′,5′-dioxo-2′,3′,4′,5′-tetrahydro-1′*H*-1,3′-bipyrrole) carboxaldehyde. The oxygen radical absorbance capacity (ORAC) was measured; however, this compound did not show any activity at 0.00–0.08 unit/µmol [42]. In 2005, from the ethanol extract of *Salvia miltiorrhiza*, five nitrogen-containing compounds, including 5-(methoxymethyl)-1*H*-pyrrole-2-carbaldehyde (**36**), were isolated and spectroscopically characterized [43]. A fused lactone derivative of Py-2-C (**48**, 3-oxo-4-benzyl-3,4-dihydro-1*H*-pyrro[2,1-*c*]oxazine-6-methylal) was isolated from the fruits of *Celastrus orbiculatus*. The structure was determined based on spectroscopic analyses [44]. From concentrated prune juice (*Prunus domestica* L. from the USA), many phenolic compounds, including a bipyrrole compound, have been isolated and characterized. Some phenolic compounds showed a high ORAC value [45]. The structure of the bipyrrole compound (**47**) had previously been identified by the same group [42]. From an extract of *Iris squria L*. (Calizona), the Py-2-C [4-(2-formyl-5-hydroxymethylpyrrol-1-yl) butyric acid **3**] was isolated along with several isoflavone derivatives [30]. A methanol extract of the leaves of *Ilex paraguariensis* contained a pyrrolezanthine derivative (**49**, in which a hydroxymethyl moiety replaced the methoxymethyl moiety of pyrrolezanthine), which was isolated and the structure was determined from the spectroscopic analyses [46] (Figure 3). All the compounds isolated from *Ilex paraguariensis* were evaluated for human neutrophil elastase inhibitory activity, however, the pyrrole derivative did not show any significant inhibitory activity. From *Lycium Chinense* fruits, three Py-2-C derivatives (**3**, **31**, and **12**) with very similar chemical structures have been isolated. The core structure was 2-formyl-5-hydroxymethyl-1-yl-pyrrol-butanoic acid (**3**), and in the derivative (**21**), the hydroxymethyl group was replaced with a methoxymethyl moiety, and in the derivative (**12**), the hydroxyl and carboxylic acid moieties were replaced with hydroxymethyl and methyl carboxylate groups, respectively [17]. A synthetic approach to these compounds has been performed using methyl (2-formyl-5-hydroxymethyl-pyrrol-1-yl)-butyrate (**50**) as a key substrate. Ten years after the first discovery of Py-2-C compounds in the fruits of *Lycium chinense*, four more Py-2-Cs (**33** and **51**–**53**) were isolated from an ethyl acetate extract of *Lycium chinense* and the structures were determined using HREIMS, ^1^H- and ^13^C-NMR, HMBC, COSY, NOESY, and HMBC. Two of the compounds were methyl propanoate derivatives, one was a dimethyl butanedioate, and one was a dimethyl pentanedioate derivative. All of these compounds have bulky *N*-alkyl side chains containing stereogenic centers (the absolute configurations were not determined) [47]. Recently, three unknown Py-2-Cs and one known Py-2-C (**54**–**57**) with short-chain carboxylic acids were reported and identified from *Lycium chinense*. The structures of these compounds were determined by spectroscopic analyses, mainly one-dimensional and two-dimensional NMR and HRMS analyses. The precise stereochemical determination was performed using the experimental electronic circular dichroism (ECD) values compared with the calculated ECD values [48]. From an ethanol extract of *Capparis spinosa*, two new pyrrole carboxaldehydes (**18** and **58**, capparisines A and B, respectively) and one furan derivative (capparisine C) were isolated, along with two known Py-2-Cs (**37** and **47**) [49]. The structures of the new compounds were determined spectroscopically, and the stereochemistry was confirmed by X-ray crystallographic analysis. All the compounds were evaluated for an inhibitory effect on apoptosis induced by Act D and TNF-α using a human hepatocyte cell line (HL-7702); however, no inhibitory effect was observed for any of the compounds. From an ethanol extract of the rhizome of *Coniogramme japonica*, six compounds were isolated and identified, including butyl 2-formyl-5-butoxymethyl-1*H*-pyrrole-1-butanoate (**59**) **[50]**. Pyrrolemarumine 4″-*O*-α-l-rhamnopyranoside (**60**) was isolated from the leaves of *Moringa oleifera*, along with eight other known compounds [51]. ^1^H-NMR spectroscopy indicated the presence of one sugar moiety, which was determined to be α-l-rhamnose from the chemical shifts of the aromatic proton and secondary methyl proton. Further acid-catalyzed hydrolysis of the compound afforded the aglycone pyrrolemarumine. From a methanol extract of *Lobelia chinensis*, 4-(2-formyl-5-butoxymethylpyrrol-1yl) butyric acid (**61**, lobechine) was isolated and the structure was determined by spectroscopic analyses [52]. From the flower buds of *Tussilago farfara*, 5-ethoxymethyl-1*H*-Py-2-C (**62**) was isolated along with a new norsesquiterpenoid (tussfarfarin A) and other known metabolites. The pyrrole derivative (**62**) was considered to be an artifact produced during the extraction process and three other new compounds were also considered artifacts [53]. A methanol extract of the roots of *Aralia continentalis* showed inhibitory activity toward protein tyrosine phosphatase 1B (PTP1B) and rat lens aldose activity (RLAR). From a hexane fraction of this methanol extract, 4-[2-formyl-5-(methoxymethyl)-1*H*-pyrrol-1-yl]butanoic acid (**21**) was isolated [54]. Two Py-2-Cs have been isolated from *Alhagi sparsifolia*, pyrrolezanthine (**10**), and a derivative of (**10**) with a methyl-ether at the 6-position of the pyrrole (**49**) [55]. From an ethanol extract of lateral root of *Aconitum carmichaelii*, two Py-2-Cs (**4** and **63**) were isolated. The structure determination of the new compounds, including **63** (aconicaramide), was performed using spectroscopic analyses. [56]. From the kernel of *Prinsepia uniflora*, two galactosides were isolated and identified as 5-[(α-d-galactopyranosyloxy)methyl]-1*H*-Py-2-C (**64**) and 6-[(α-d-galactopyranosyloxy)methyl]-3-pyridinol based on the spectroscopic analyses [57]. From an ethanol extract of the root of *Aconitum flavum* Hand,-Mazz., 4-(2-formyl-5-hydroxymethylpyrrol-1-yl)butyric acid (**4**) was isolated, along with another 14 compounds [58]. From the rhizomes of *Aristolochia fordiana*, a new alkaloid glycoside, fordianoside, was isolated and structurally determined. In the separation procedure, the known pyrrole alkaloid, 4-[formyl-5-(hydroxymethyl)-1*H*-pyrrol-1-yl]butanoic acid (**3**) was also isolated [59]. The macrophage-activating constituents of *Morus alba* fruits were purified, and the structures of the isolated compounds were determined based on the spectroscopic analyses. The five isolated compounds were Py-2-Cs, an *N*-unsubstituted-Py-2-C bearing a hydroxymethyl group at the 5-position (**4**) and four Py-2-C-1-butanoic acid derivatives, non-substituted (**65**), and compounds with hydroxymethyl (**3**) or methoxymethyl (**21**) groups at the 5-position. These five compounds had not been previously isolated from *Maorus* sp., and the branched alkoxy derivative of pyrrole-2-carbxaldehyde (**66**, morrole A) was a new compound [60]. The same group isolated 17 other pyrrole alkaloids from the fruits of *Morus alba*, and five new compounds were characterized spectroscopically and named morroles B–F (**67**–**71**). All 17 of these compounds had a formyl substituent at the 2-position of the pyrrole ring; therefore, these compounds can also be classified as Py-2-C derivatives [58]. Py-2-Cs have also been isolated and identified from the seeds of very common fruit, such as watermelon (*Citrullus lanatus*). Two new pyrrole carboxaldehydes—1-[5-(5-hydroxymethyl-1*H*-Py-2-C-1-yl)ethyl]-1*H*-pyrazole (**72**) and 1-({[5-(α-d-galactopyranosyloxy)methyl]-1*H*-pyrrole-2-caboxaldehyde-1-yl}-ethyl)-1*H*-pyrrazole (**73**)—have the *N*-positions of a pyrazole ring and the pyrrole ring connected by an ethylene chain [61]. This type of compound had not previously been reported in the literature, although pyrrole–imidazole alkaloids have been reported [62]. The structures of these compounds were determined using ^1^H- and ^13^C-NMR spectroscopic techniques. The α-configuration of the sugar moiety at the 1-position was established based on the coupling constant and was also confirmed by acid hydrolysis and subsequent derivatization using l-cysteine methyl ester hydrochloride and *o*-tolylisothiocyanate (Figure 3).

From the fruit of *Capparis pinosa*, two new Py-2-C derivatives—capparisine A (**18**) and capparisine B (**58**)—were isolated, along with the known Py-2-C derivatives 2-(5-hydroxymethyl-2-formyl-1-yl) α-methyl-acetic acid lactone (**37**) and *N*-(3′-maleimidyl)-5-hydroxymethyl-2-pyrrole formaldehyde] (**47**) [49]. From bee-collected *Brassica campestris* pollen, pollenpyrrosides A (**17**) and B (**18**) were isolated, and the precise structures were determined using several spectroscopic analysis techniques (NMR, UV, IR, MS, and X-ray) and chemical evidence [63]. Pollenpyrroside A and caparisine B are diastereomers of the same spiroketal alkaloid. The total synthesis of these compounds has been reported, which started with d-fructose as a chiral substrate followed by microwave-assisted bishydroxymethylation of the pyrrole ring and a key acid-catalyzed cyclization [64]. Another approach used a Maillard-type condensation of an amine derived from deoxy-d-ribose. After several steps, pollenpyrroside A (**17**) was synthesized and the exact stereochemistry of **17** was confirmed [65]. Similar spiroketal pyrrole alkaloids, acrotartarins A and B, were also isolated from *Acorus tartainowii*. The structures (absolute configurations) of acrotartarins A and B were determined using various spectroscopic analyses and X-ray diffraction analysis [66]. After the isolation of acortatarins A (**18**) and B (**74**), total synthesis has been attempted by several groups, and the absolute configurations of acortatarins A and B were determined [67,68,69,70,71]. From the flower buds of the daylily, the glutamine derivatives hemerocallisamine I (**75**) and hemerocallisamine II (**76**) possessing Py-2-C moieties have been isolated and characterized based on the spectral evidence (UV, MS, HRMS, and ^1^H- and ^13^C-NMR) and X-ray analyses [72,73]. The total synthesis of hemerocallisamine I using a Maillard-type condensation of a *cis*-4-hydroxy-l-proline derivative with a dihydroxypyranone is a key step, which afforded the 2-formylpyrrole ring system. Subsequent removal of the protecting group gave hemerohallisaline I and the optical rotation data indicated the stereochemistry of hemerohallisaline I (revised as 2*S*,4*S*-hemerocallisamine I) [74]. From the traditional Chinese medicine (TCM) anti-arrhythmic formula Shensong Yangxin capsule, four Py-2-Cs (shensongines A (**19**), B (**20**), and C (**77**), and pollenpyrroside B (**18**)) bearing a morpholino chromophore (spiroalkaloids) have been isolated and characterized based on the spectroscopic data (^1^H- and ^13^C-NMR and HMBC) [75]. The configurations of these compounds were determined by comparison with the known literature values for the optical rotation and the ^1^H-NMR spectra, including ^1^H-^1^H COSY and HMBC. Some of these compounds were already known: shensongine A is xylapyrroside A (*ent*-(enantiomer)-capparisine B), shensongine B is xylapyrroside B, and pollenpyrroside B has the same structure as ascrotatarin A (*ent*-(enantiomer) capparisin A). The new Py-2-C, methyl 2-[2-formyl-5-(hydroxymethyl)-1*H*-pyrrol-1-yl]propanoate (**78**), was isolated from a Goji berry-contaminated commercial sample of African mango, along with three other known Py-2-Cs (**22**, **4**, and **37**) [76]. From an ethanol extract of the roots of *Ranunculus ternatus* Thunb (20 kg) four Py-2-Cs (**79**, **80**, **81**, and **82**; 15, 12, 10, and 8 mg, respectively) were isolated that bear monosaccharides or disaccharides at the hydroxymethyl group on the pyrrole ring. The structures of these compounds (named ranunculins A–D) were elucidated by spectroscopic analyses and the calculated ECD values [77]. These molecules interestingly have a *tail-to-tail* bond (6,6 ether linkage). This is a previously unreported hybrid of a γ-amino acid and a sugar. New heterocyclic compounds have also been isolated and characterized from *Ranunculus ternatus* Thunb., including the furfural derivative 4-{2-[(2*S*-2,3-dihydroxypropoxy)methyl]-5-formyl-1*H*-pyrrol-1-yl}butanoic acid (**83**) [78].

From the leaves of *Nicotiana tabacum*, the pyrrolezanthine (**10**) was isolated along with a benzofuran derivative (3-acetyl-7-hydroxy-6-methoxy-2-methylbenzofuran-4-carboxylate) [79]. The above data show that Py-2-Cs have been isolated from many fungal and plant sources. In some cases, the same compound has been isolated from different sources and named differently. For example, shensongine A is the same compound as xylapyrroside A (*ent*-capprisine B), acortatarin A is the same as pollempyrroside B (*ent*-capparisine A), and shensongine B is the same as xylapyrroside B. All of these compounds have a pyrrolomorpholine spiroketal structure and chiral carbon atoms [80]. In some cases, a stereochemical revision has been performed after the isolation and original characterization. Some of the plant sources of Py-2-Cs are TCMs that are used for the treatment of various diseases, including cancer and lifestyle diseases. Thus, the isolation and identification of the key materials in the plant sources of Py-2-Cs is important. Any active compounds should also be efficiently synthesized. From an ethanol extract of the flower of *Junglans regia*, seven new and four known alkaloids were isolated. Among the 11 isolated alkaloids, the two Py-2-Cs (5,6,11,12-tetrahydropyrrolo[1′,2′:1,2]azepino[4,5-*b*]indole-3-carboxaldehyde (**96**) and 5-(ethoxymethyl-1-(4-hydroxyphenethyl)-1*H*-Py-2-C (**97**) were isolated and the structures were elucidated by spectroscopic analyses [81] (Figure 4).

Recently, from the root of *Reynoutria ciliinervis* (Nakai) *moldenke*, a new derivative of pyrrolezanthine (5-(butoxymethyl)-1-(4-hydroxyphenyl)-1*H*-pyrrole-2-carbozaldehyde, (**86**)) was isolated and the structure was determined from spectroscopic analyses (NMR and HR-ESI-MS) along with single-crystal X-ray diffraction data. The structure of (**86**) was the same as that of the pyrrolezanthine (**10**), except for a butoxymethyl substituent in place of the hydroxymethyl in the pyrrolezanthine (**10**) at the 5-position on the pyrrole ring. A spectral comparison was performed with the pyrrolezanthine (**10**) [82]. Eleven Py-2-Cs have been isolated from the roots of *Angelica dahurica*, six new compounds, dahurines A–F (**87**–**92**), and five other Py-2-Cs (**59**, **77**, **93**, **94**, and **95**) [83]. Two compounds had already been synthesized; however, these two compounds (**93** and **94**) had not previously been isolated from a natural source. The structure determination of these compounds was based on spectroscopic data (1D and 2D NMR, IR, and HRESIMS) and ECD methods. The root of the medicinal plant *Angelica dahurica* is commonly used in Eastern countries and contains various types of compounds which have been summarized in recent literature [84]. From an *n*-butanol extract of *Selaginella delicatula,* eight metabolites were isolated, including the new pyrrole alkaloid butyl 2-formyl-5-hydroxymethyl-1*H*-pyrrole butanoate (**95**), and the structure of (**95**) was determined based on the spectroscopic analyses [85]. From *Hosta plantaginea*, five new compounds, named hostines A–E, were isolated and characterized, including a new carboxaldehyde (5-ethoxymethyl-pyrrolemarumine, **96**) [86]. From *Urticae Fissae Herba,* thirteen alkaloids, two lignins, and three amides were isolated. The thirteen alkaloids included nine Py-2-Cs: seven known (**12**, **21**, **18**, **58**, **59**, **61**, and **97**) and two unknown Py-2-Cs. The structures of the unknown Py-2-Cs (**93** (dahurine F) and **98**) were determined based on the spectroscopic analyses [87]. From *Moringa oleifera* seeds, seven new Py-2-Cs (**99**–**105**) and four known Py-2-Cs (**5**, **10**, **96**, and **106**) were isolated and the structures of the new compounds were characterized mainly based on NMR spectroscopic analyses and HR-ESI-MS [88] (Figure 4). 

### 3.3. Pyrrole-2-Carbaldehydes from Microorganisms

Py-2-Cs have also been isolated from sources other than plants and fungi. In 1975, the isolation and characterization of several saturated and unsaturated 3-alkyl-Py-2-C derivatives (saturated (**107**–**111**), mono-unsaturated (**112**–**113**), and di-unsaturated (**114**)) have been reported from the marine sponge *Oscarella lobularis* [89]. The structure of these compounds was determined mainly from NMR and MS spectroscopic studies. The position and number of the double bonds were determined from the chemical reaction with mercury acetate. In 1980, from the marine sponge *Laxosuberites* sp. (collected from Canon Atoll), six 5-substituted-Py-2-Cs (**115**–**120**) were isolated and characterized [90]. In 1997, from the sponge *Mycale micracanthoxea*, two Py-2-Cs, mycalazal 1 (**121**) and mycalazal 2 (**122**), were isolated and characterized from spectroscopic analyses, along with twelve 5-acyl-2-hydroxymethylpyrroles (^1^H- and ^13^C-NMR, COSY, DEPT, IR, and MS) [91]. Mycalazal 2 (**122**) is a dihydro derivative of mycalazal 1 (**121**). Thirteen new Py-2-C derivatives along with two known Py-2-Cs (**115** and **116**) were isolated from *Mycale microsigmatosa* and *Desmapsamma anchorata* from the Caribbean Sea (Venezuela). These thirteen pyrrole carboxaldehydes were categorized into four groups. The groups were pyrrole-2-carboxaldehydes bearing either a nitrile group (**123**), a linear carbon chain (**124** and **125**), a branched carbon chain (**126** and **127**), or an unsaturated carbon chain (**128**–**135**). All of these Py-2-Cs have a long carbon chain (15–23 carbon atoms, branched and non-branched) at the 5-position of the pyrrole ring and one compound has a functionalized nitrile group at the end of the alkyl chain [92]. This nitrile functionalization has also been found in compounds isolated from *Laxosuberites* sp. (**119**). From *Mycale mytilorum*, four new and two known compounds were isolated. Among the four new compounds, two Py-2-Cs, 5-octadecyl Py-2-C (**136**) and (6′*Z*)-5-(6′-hexeicosenyl) Py-2-C (**137**), were spectroscopically identified [93]. The structures of these compounds were determined based on spectroscopic analyses (HRMS, ^1^H- and ^13^C-NMR, UV, and IR) (Figure 5). In 2000, 5-alkyl-2-formylpyrrole derivatives were isolated from the sponge *Mycale tenuispiculta* [94]. The functionalization at the end of the alkyl chain was dependent on the chain length. The moieties at the ends of the alkyl chains are nitrile (**138**), oxime (**139**, mycaleoxime), and methyl (**140**) groups. These differences in the functional groups might affect the physiological activities of these compounds. The stereochemistry is also interesting, the Py-2-Cs bearing nitrile or *N*-oxime groups at the termini were cis-alkenes, and the Py-2-C bearing a methyl group at the terminal was a trans-alkene. The carbon chain lengths of these compounds are all long but not the same length, which suggests that different metabolic processes are responsible for the formation of these compounds. From the sponge *Mycale cecilia*, 14 new (**141**–**154**) and eight known (**115**, **116**, **109**, **112**, **123**, **126**, **127**, **131**) Py-2-Cs were isolated and the structures were determined from NMR and MS analyses (the regiochemistry of **120** and **123** was revised from the original work (93)). These compounds were categorized into six groups: Py-2-Cs bearing a triene moiety in the side chain (**141**–**143**); a diene moiety in the side chain (**144**–**146**); a single double bond in the side chain (**147**–**150**, **112**, and **118**); having a branched alkyl side chain (**151**, **112**, and **127**); having a linear alkyl side chain (**115**, **116**, and **109**); and bearing a nitrile group in the side chain (**152**–**154** and **123**) [95]. The number of unsaturated carbon bonds (0, 1, 2, 3, 5, and 6) is also different in these compounds. From the marine sponge *Mycale* sp. (from Turtle Bas, Palau) 18 new lipophilic Py-2-Cs (**155**–**172**) and eight known Py-2-Cs (**115**, **116**, **123**, **126**, **127**, **141**, **152**, and **153**) were isolated and spectroscopically characterized [96]. These compounds were also categorized into three groups: (1) with nitrile substitution at the terminus of the carbon chain (1, diene chain; 2, ene chain; 3, saturated carbon chain); (2) with an unsaturated carbon chain (1, triene; 2, diene; 3, ene); and (3) with a saturated carbon chain (1, linear carbon chain; 2, branched carbon chain). Some of these compounds were shown to inhibit hypoxia-inducible factor-1 (HIF-1) activation. The mechanism of action of these compounds is believed to be the disruption of mitochondrial ROS-regulated HIF-1 signaling [97]. From the Indonesian marine sponge *Mycale phyllophia*, two Py-2-Cs were isolated in a 2:1 mixture: 5-pentadecyl-1H-Py-2-C (**115**) and (6′*E*)-5-(6′-pentadecenyl)-1*H*-Py-2-C (**173**). The E-isomer was isolated and characterized based on the spectroscopic analyses (^1^H- and ^13^C-NMR and MS) [97]. Explant cultures of *Mycale Cecilia* were carried out using two methods (natural and closed conditions). The final survival of *Mycale cecilia* was higher in the natural environment (95% ± 7.07%) than in the enclosed system (65% ± 21.21%). The growth was also higher in the natural environment (207%) than in the enclosed system (65%). The presence of Py-2-Cs, including mycalazals and mycalenitrile-related compounds, was determined from HPLC and the ^1^H NMR spectra. The concentration of Py-2-Cs (in the dry weight extracts of the sponge) isolated from the sponge *Mycale cecilia* at the start and end of the experiments was 0.13% ± 0.0006% and 0.35% ± 0.01%, respectively. The concentration of Py-2-Cs isolated in the natural and enclosed systems was 0.24% ± 0.006% and 0.25% ± 0.01%, respectively [98]. Some mycalazal Py-2-Cs (**4**, **7**, **8**, and **10**) were identified based on the HPLC retention times. Thus, the establishment of a culture system enabled the easy isolation of the amounts of Py-2-Cs required to examine the biological activities of the compounds. From the South China Sea sponge *Mycale lissochela*, two new Py-2-Cs (**174** and **175**) having lipophilic side chains with one or two unsaturated carbon bonds and a nitrile group at the terminus were isolated, along with the five known Py-2-Cs (**123**, **138**, **153**, **154**, and **164**) [99]. Micalenitrile-15 (**174**) has two double bonds (cisoid conformation) and a nitrile functional group at the terminus of the lipophilic alkyl chain. Micalenitrile-16 (**175**) has one double bond (cisoid) and a nitrile group at the end of the alkyl chain. From the mangrove-derived actinomycete *Jishengella endophytica* 161111, a new pyrazine alkaloid was isolated and characterized. Along with this new compound, (*S*)-4-isobutyl-3-oxo-3,4-dihydro-1*H*-pyrrolo[2,1-*c*][1,4]oxazine-6-carbaldehyde (**39**), (*S*)-4-isopropyl-3-oxo-3,4-dihydro-1*H*-pyrrolo[2,1-*c*][1,4]oxazine-6-carbaldehyde (**38**), (4*S*)-4-(2-methylbutyl)-3-oxo-3,4-dihydro-1*H*-pyrrolo[2,1-*c*][1,4]oxazine-6-carbaldehyde (**40**), (*S*)-4-benzyl-3-oxo-3,4-dihydro-1*H*-pyrrolo[2,1-*c*][1,4]oxazine-6-carbaldehyde (**49**), and 5-(methoxymethyl)-1*H*-pyrrazole-2-carbaldehyde (**36**) were isolated and characterized [100]. From the liquid culture medium (supplemented rice with liquid ISP-2), two known pyrrole alkaloids, (*S*)-4-benzyl-3-oxo-3,4-dihydro-1*H*-pyrro[2,1-*c*][1,4]oxazine-6-carbaldehyde (**48**) and (*S*)-4-isobutyl-3-oxo-3,4-dihydro-1*H*-pyrro[2,1-*c*][1,4]oxazine-6-carbaldehyde (**39**) were isolated, along with diketopiperazine derivatives [101]. The meroterpenoid derivatives cinerols A–K were isolated from the marine sponge *Dysidea cinera* collected from the South China Sea [102]. Among these metabolites, the Py-2-C-containing terpenoid cinerol I (**176**) was isolated and characterized via spectroscopic analyses (HRESIMS and ^1^H- and ^13^C-NMR) (Figure 5).

The sponge *Mycale* sp. has produced a variety of Py-2-Cs along with other metabolites, which have been summarized in recent review articles [103,104]. 

In 2014, seven new compounds, including five new Py-2-Cs (**177**–**181**, jiangrines A–E, along with the pyrrolezanthine, **10**) were isolated from an Actinobacterium, *Jiangella gansuensis*. The structures of the compounds were determined mainly based on NMR spectroscopic techniques (COSY, HMBC, and NOE,) and HRESIMS. The precise stereochemical configuration of the glycerol moiety at the 5-position of the pyrrole ring was determined based on the chemical reaction in which the hydroxyl group at the 2-position of glycerol attacks the carbonyl moiety of the aldehyde to form a seven-membered cyclic hemiacetal structure in methanol. From NOE correlation studies, the stereochemical configuration of the glycerol moiety of jiangrine A was determined to be *S* (for the carbon attached to the ring) and *R* (for the carbon adjacent to the carbon attached to the ring). From a comparison to the chemical shifts of jiangrine A, the stereochemistries of jiangrines B, C, and D were determined. Under the reaction conditions, the separation of the stereoisomers of jiangrine C and jiangrine D was not achieved. These two compounds were present as a 1:1 mixture and this mixture was used in the biological experiments [105]. A new Py-2-C (**182**, jiangrine G), along with the known Py-2-Cs jiangrine A (**183**, revised structure of **117**) and pyrrolezanthine (**10**) were isolated from the fermentation broth of *Jiangella alba* associated with the traditional Chinese medicinal plant *Maytenus austroyunnanensis*. The structure of jiangrine G is very similar to jiangrine A, including the stereochemistry of the moiety at the 3-position, except for the hydroxyl moiety on the phenyl ring attached at the 1-position of the pyrrole ring is absent in jiangrine G. The precise determination of the structure of jiangrine G was determined from spectroscopic studies (^1^H- and ^13^C-NMR, HRESIMS, and CD) [106] (Figure 6).

## 4. Physiological Activities of Pyrrole-2-carbaldehydes

### 4.1. Physiological Activities of Pyrrole-2-carbaldehydes Originating from Fungi

From the extract of *Mycoleptodonoides aitchisonii,* 11 compounds were isolated, including a Py-2-C (**9**). All the compounds were examined for NAD(P)H quinone oxidoreductase (NQO1) activity. The Py-2-C (**9**, 100 µg/mL) showed strong NQO1 inducing activity [12] (Table 1).

A Py-2-C derivative (100 µg/mL) derived from makomotake (*Zizania latifolia*) showed moderate NQO1 induction activity in a Hepa 1c1c7 cell line and no cytotoxic effect was observed under the reaction conditions. The three Py-2-Cs (**12**–**14**) have been subjected to some bioassays, including the suppression of osteoclast formation (which has been reported for makomotine A); however, no significant biological activity has been observed for these compounds [18] (Table 1).

Administration of a *Cordyceps millitaris* extract (100 mg/kg, 300 mg/kg body weight) reduced the body weight gain and food efficiency ratio in C58BL/6J mice fed a high-fat diet. Among the compounds isolated from *Cordyceps millitaris*, cordyrrole A (**16**) showed a significant reduction in adipocyte differentiation and pancreatic lipase activity in 3T3-L1 cells [20] (Table 1).

Four Py-2-Cs (**17**–**20**) isolated from *Xylaria nigripes* and structurally related synthetic derivatives have shown moderate to strong antioxidant effects. Rat A7r5 vascular smooth muscle cells were subjected to *t*-butyl hydroperoxide-induced oxidative stress after pretreatment for 4 h with Py-2-Cs (**17**, **18**, and **20**) at varying concentrations. The administration of Py-2-Cs attenuated cell death in a concentration-dependent manner. Among these compounds, **20** showed the strongest activity and a synthetic diastereomer of **17** also showed similar strong activity [21] (Table 1).

From the edible mushroom *Phlebopus portentosus*, seven Py-2-Cs (**29**–**35**) were isolated and characterized. SH-SY5Y cells were pretreated with each of these compounds or *N*-acetylcysteine (NAC) at 10 μM for 2 h prior to treatment with H_2_O_2_ (10 μM). The Py-2-C with an ethylene bridge between the 3-position of the indole ring and the *N*-position of the pyrrole carboxaldehyde (**35**) showed the strongest neuroprotective activity, which was almost the same as NAC. The improvement in the cell viability for NAC and **36** was 24.5% and 26.5%, respectively. The other compounds showed moderate to mild activity (5.8%–15.7%). None of these compounds showed significant acetylcholine esterase inhibitory activity under the reaction conditions [26] (Table 1).

Among seven metabolites isolated from *Purpureocillium lavendulum,* 5-methoxymethyl-1*H*-Py-2-C (**37**) showed the strongest toxicity toward *M. inocognita*. The administration of 5-methoxymethyl-1*H*-Py-2-C (**37**) at 400 ppm enhanced the nematocidal activity with increasing time. The nematocidal activity of 5-methoxymethyl-1*H*-Py-2-C (**37**) was 23.20 ± 2.33 (12 h), 29.30 ± 2.33 (24 h), 41.60 ± 3.80 (48 h), 68.87 ± 3.63 (72 h), and 98.23 ± 0.81 (96 h), while that of the control was 2.06 ± 0.22 (12 h), 2.07 ± 0.12 (24 h), 2.67 ± 0.58 (48 h), 4.40 ± 0.56 (72 h), and 5.73 ± 1.10 (96 h) (adjusted mortality rate > 90%). Similar observations were also found for the egg hatching rate of *M. incognita* [107] (Table 1). 

### 4.2. Physiological Activities of Pyrrole-2-carbaldehydes Originating from Plants

The anti-platelet aggregation activity of pyrrolezanthine (**10**) was examined along with other metabolites. Some metabolites showed anti-platelet activity but **10** did not show any activity [14]. The antioxidative activity of magnolamide (**44**) and a structurally related analog of magnolamide was investigated in the copper-induced oxidation of freshly prepared human LDL lipids by measuring the absorbance at 234 nm of the conjugated dienes in the presence or absence of Py-2-Cs. The IC_50_ values of magnolamide (**44**) and the structurally related analog were 9.7 ± 2.8 and 16.9 ± 2.3, respectively. The IC_50_ values of resveratrol and probucol under the same conditions were 13.1 ± 2.6 and 8.7 ± 1.4, respectively. These results showed that mangnolamide (**44**) had almost the same inhibitory activity against Cu^+^-induced LDL oxidation as probucol and resveratrol [108].

Pyrrolezanthine-6-methyl ether (**49**) and other metabolites (17 compounds) have been isolated from a methanol extract of the leaves of *Ilex paraguariensis* and were investigated in a human neutrophil elastase (HNE) inhibitory assay. Some compounds showed significant HNE inhibitory activity; however, **49** did not show any inhibition (the IC_50_ value was not measurable) [46] (Table 2).

Three compounds (**3**, **21**, and **12**) isolated from *Lycium chinense* fruits and the synthetic precursor (**50**) of **12** showed high hepatoprotective activity at 0.1 μM concentrations [17]. Silybin (the positive control) showed the highest hepatoprotective activity (45.5%, 50 μM), and the compounds **3**, **52**, and **53** showed protective activity of 64.4% ± 3.9%, 65.5% ± 5.6%, and 38.5% ± 4.9%, respectively, at 0.1 μM. In general, the hepatoprotective effect was higher than that of silybin; however, at high concentrations such as 10 μM, the hepatoprotective effect of compounds **3**, **52**, and **53** decreased to 3.2% ± 1.0%, 14.2% ± 2.2%, and 15.9% ± 1.7%, respectively. The butanoic acid derivatives **3** and **21** showed higher hepatoprotective activity than the butanoic acid esters **12** and **50** at low concentrations (0.1 μM) (Table 2).

Forty-six compounds were characterized from a methanol extract of *Lobelia chinensis*, including the Py-2-C lobechine (**61**). Although **61** showed no inhibitory activity against HSV-1 replication or inhibition of superoxide generation by human neutrophils in response to *N*-formyl-methionyl-leucyl phenylalanine/cytochalasin B, **61** showed moderate elastase release inhibitory activity. The IC_50_ value of **66** for elastase release inhibition was 25.01 ± 6.95 μM (the IC_50_ value of the positive control LY294002 was 2.64 ± 0.29 μM) [52].

From a methanol extract of the roots of *Aralia continentalis*, 18 compounds were isolated and characterized based on the spectroscopic analyses, including the pyrrole-2-carabozaldehyde (**21**, 4-[2-formyl-5-(methoxymethyl)-1*H*-pyrrol-1-yl]butanoic acid). The inhibitory activity of **68** against rat lens aldose reductase (RLAR) showed moderate inhibitory activity [54]. The IC_50_ value of **21** was 39.71 ± 1.77 μM (the IC_50_ value of the positive control quercetin was 3.92 ± 0.96 μM) (Table 2).

Five Py-2-Cs (**3**, **4**, **21**, **65**, and morrole A (**66**)) isolated from *Morus alba* fruits were subjected to biological assays using RAW 264.7 cells. By measuring the production of nitric oxide (NO), TNF-α, and IL12, two butanoic acid derivatives bearing hydroxymethyl (**3**) or methoxymethyl (**21**) substituents at the 5-position of the pyrrole ring were shown to cause significant macrophage activation and also stimulated phagocytic activity in RAW 264.7 cells. Three other Py-2-Cs (**4**, **65**, and **66**) did not show significant activity [60]. Morrole A (**66**), which has the bulky substituent CH(CH_3_)-CH(CH_3_)-OH on the hydroxymethyl moiety did not show significant activity in any of the assays. The less bulky CH_3_ derivative showed significant activity, which suggested the importance of the influence of steric hindrance on the activity (Table 2).

From an ethanol extract of the lateral root of *Aconitum carmichaelii*, two pyrrole carbaldehydes (**4** and **63**) were isolated along with C_20_-diterpenoid derivatives. Aconicaramide (*N*-(l-prolyl)-5-hydroxymethyl-1*H*-pyrrole-2-carbaldehyde, **63**) showed moderate antibacterial activity against *M. caseolyticus*, *S. epidermidis*, and *S. aureus* (the MIC values were 200, 400, and 800 μg/mL, respectively) [56] (Table 2).

Twelve Py-2-Cs (**12**, **21**, **23**, **24**, **32**, **37**, **38**, **39**, **40**, **47**, **48**, and **50**) and five pyrrole-carboxaldehydes (**67**, **68**, **69**, **70**, and **71**, named morroles B, C, D, E, and F, respectively) were tested for inhibitory activity against pancreatic lipase activity using porcine pancreatic lipase (in vitro). Methyl 2-[2-formyl-5-(methoxymethyl)-1*H*-pyrrol-1-yl] propanoate (**32**) and 2-(5′-hydroxymethyl-2′-formylpyrrol-1′yl)-3-(4-hydroxyphenyl)-propionic acid lactone (**23**) showed significant inhibitory activities of 40% and 70% at 100 μM, respectively [109]. Of the structurally related compounds (**67**, **68**, and **32**), only **32**, bearing methoxymethyl and ester substituents, showed moderate lipase inhibitory activity. This result suggested the importance of the hydrophobicity of the compound for lipase inhibitory activity. Of the pyrrole lactone derivatives (**23** and **48**), only **23** showed lipase inhibitory activity. The difference between **23** and **48** is the presence of a hydroxy group at the para position of the substituted benzyl group in **23** (Table 2).

(1-[5-(5-Hydroxymethyl-1*H*-Py-2-C-1-yl)ethyl]-1*H*-pyrazole (**72**) showed moderate inhibitory activity toward melanogenesis (induced by α-melanocyte stimulating hormone in the B16 melanoma 4A5 cell line) without cytotoxicity (80.3% ± 9.3% at 30 μM and 64.1% ± 5.8% at 100 μM), which was comparable to the values for arbutin (the positive control) in in vitro experiments. The glycosylated pyrazole derivative (**73**) also showed inhibitory activity (86.0% ± 2.4% at 30 μM and 48.6% ± 0.6% at 100 μM); however, **73** showed cytotoxicity at 100 μM (cell viability 70.4% ± 1.9% [61] (Table 2).

From *Capparis spinosa*, five alkaloids, including two new and two known Py-2-Cs (**18**, **37**, **47**, and **58**), were isolated. The two new Py-2-Cs were named capparisine A (**18**) and capparisine B (**58**). The inhibitory effects of these compounds on the apoptosis (HL-7702 human hepatocyte cell line) induced by Act D (200 ng/mL) and TNF-α (2 ng/mL) were examined by evaluating the nuclei size and the average number of mitochondrial masses. None of these compounds showed any apoptosis inhibitory activity on the human hepatocyte cell line HL-7702 [49] (Table 2).

The pollenopyrrosides A (**17**) and B (**18**) were assayed using various cancer cell lines including A549 (human alveolar adenocarcinoma), Bel7420 (human hepatoma), BGC-823 (human gastric cancer), HCT-8 (human intestinal adenocarcinoma), and A2780 (human ovarian cancer) at 10 μg/mL concentrations. At the experimental concentration range, neither of these two compounds showed any cytotoxic effect on these cancer cell lines [63].

Acortatarins A (**18**) and B (**74**) showed inhibition of ROS generation induced in high-glucose-stimulated mesangial cells. In a fluorescent assay using 2′,7′-dichlorofluorescein, **18** and **74** showed the highest inhibition activity (statistically meaningful) at 10 and 50 μM, respectively. After mesangial cells were pretreated with **18** for 1 h, the cells were incubated with a high concentration of glucose for 1–24 h. The fluorescent intensity showed a decrease (ca. 50% compared with the high glucose-treated cells) in the pretreated cells at various times (1, 3, 6, 12, and 24 h) [66]. The morpholine motif may plausibly be in part responsible for these results (Table 2).

The hemerocallisamine derivatives (**75** and **76**) isolated from the daylily [69,70] were tested for their inhibitory activity against 42-mer amyloid β-protein aggregation and the effect on nerve growth factor using PC12 cell lines. Hemellocallisamine I (**75**) did not show any significant effects in either test [73] (Table 2).

Shensongines A (**19**) and C (**77**), isolated from an anti-arrhythmic TCM, showed cardiovascular activity by shortening the action potential duration in rat myocardial cell lines at 10^–6^ mol/L [75]. Shensongine B (**20**) and pollenopyrroside B (**18**) did not show any cardiovascular activity. Shensongine A and shensongine C might affect ion channels, i.e., inhibit L-type calcium channel opening or facilitate potassium channel opening.

The new Py-2-C, methyl 2-[2-formyl-5-(hydroxymethyl)-1*H*-pyrrol-1-yl]propanoate (**78**), and three known Py-2-Cs (**3**, **4**, and **37**) isolated from a Goji berry-contaminated commercial sample of African mango were tested for hydroxyl radical scavenging activity and quinone-reductase (QR) induction activity. The *N*-substituted pyrrole-2-caroboxaldehydes (**78** and **3**) showed hydroxyl radical scavenging activity (ED_50_ = 16.7 and 11.9 μM for **3** and **78**, respectively) and QR induction activity (CD = 43.1 and 4 μM for **78** and **3**, respectively). The *N*-unsubstituted Py-2-Cs (**4** and **37**) did not show hydroxyl radical scavenging or QR induction activity. These results suggested the importance of the *N*-substituent for the activity. Furthermore, the butanoic acid derivative **3** showed higher QR induction activity than **78**, which suggests the importance of the influence of the chain length at the *N*-position on the activity [76] (Table 2).

Pyrrolezanthine (**10**) isolated from *Nicotiana tabacum* showed moderate cytotoxicity against lung cancer A-549 and human colon cancer SW480 cell lines with IC_50_ values of 38.3 and 33.7 μM, respectively [79].

All the isolated compounds (including separated enantiomers) from the flowers of *Juglans regia* were evaluated for growth inhibitory activity against various cancer cell lines using the MTT assay, with taxol and NK109 as positive controls. Only 5,6,11,12-tetrahydropyrrolo[1′,2′:1,2]azepino[4,5-*b*]indole-3-carboxaldehyde (**84**) showed significant inhibitory activity against the cancer cell lines. The IC_50_ values of **84** against human colorectal carcinoma (HCT-116), human hepatocellular carcinoma (HepG2), human gastric adenocarcinoma (BGC-823), human non-small-cell lung carcinoma (NCI-H1650), and human ovarian carcinoma (A2780) cells were 2.87, 1.87, 2.28. 2.86, and 0.96 μM, respectively. The IC_50_ values of NK109 (1-hydroxy-2-methoxy-12-methyl-[1,3]-dioxola[4′,5′;4,5]-benzo[1,2-*c*]phenanthridine-12-ium chloride) in the same cancer cell lines were 3.94, 1.19, 1.84, 1.75, and 0.90 μM, respectively. The other compounds, including 5-(ethoxymethyl)-1-(4-hydroxyphenethyl)1*H*-pyrrole-2-carbaldehyde (**85**), showed IC_50_ values > 10 μM [81].

The antimicrobial activity of 5-butoxymethyl-1-(4-hydroxyphenethy)-1*H*-Py-2-C (**86**) isolated from *Reynoutria ciliinervis* (Nakai) moldenke was examined against 11 bacteria using the antibiotics streptomycin sulphate, penicillin sodium, and carbendazim as positive controls. The MIC values of **86** indicated that **86** had weak to moderate antimicrobial activity against 10 strains (*E. coli*, *P. aeruginosa*, *S. aureus*, *S. lactis*, *C. mandshurica*, *G. saubinerii*, *S. turcica*, *A. alternate*, *B. cinerrea*, and *P. anthracnose*), and **86** showed potent antifungal activity against *Sclerotinia sclerotiorum* (MIC = 31.2 μg/mL). This MIC value was the same as the value for carbendazim (a positive control) [82] (Table 2).

Eleven Py-2-Cs have been isolated and characterized from the roots of *Angelica dahurica*. Six new pyrrole-2-caroboxides (**87**–**92** named dahurines A–F) and another five known Py-2-Cs (**93**, **94**, **95**, **59**, and **75**) were evaluated for inhibitory activity toward acetylcholine esterase (AChE) using Ellman’s method. Five compounds showed weak inhibitory activity (IC_50_ = 47.5–52.5 μM), and six compounds did not show any inhibitory activity (<50% inhibition at 100 μM). Among the new compounds, dahurines B (**88**), C (**89**), and D (**90**) and compounds **59** and **75** showed weak inhibitory activity (IC_50_ = 52.0 ± 0.5, 48.2 ± 0.1, 47.5 ± 0.2, 50.4 ± 0.6, and 52.5 ± 3.4 μM, respectively, IC_50_ = 0.6 ± 0.1 μM for the positive control huperzine A). The other compounds (**87**, **91**–**94**, and **95**) did not show any inhibitory activity (<50% inhibition at 100 μM) [83] (Table 2).

The inhibitory activities of **95** (isolated from *Selaginella delicatula*) and delicatulines were evaluated against HBV surface antigen and HBV DNA in HepAD38 cells; however, all of these compounds showed very weak or no inhibitory activity against HBV [85].

The compound 5-ethoxymethyl-pyrrolemarumine (**96**), isolated from *Hosta plantaginea*, showed significant anti-inflammatory activity against LPS-induced stress. The IC_50_ value was 8.6 ± 0.7 μM, which was very close to the value for the positive control parthenolide (4.96 ± 0.5 μM). Analysis of the mechanism of action revealed an interaction of 108 with *i*NOS protein [86] (Table 2).

Various compounds, including 10 Py-2-Cs (**12**, **16**, **21**, **18**, **58**, **59**, **61**, **92**, **97**, and **98**) have been isolated from *Uriticae Fissae Herba* and evaluated for analgesic activity in mice. Four Py-2-Cs (**59**, **61**, **92**, and **98**) significantly decreased the number of writhes after injection of the compound (2 mg/BW) via the tail vein [87]. A similar result was observed in a hot-plate experiment; a posterior injection of the compounds (2 mg/BW) significantly increased the pain threshold of the mice on the hot-plate, especially for compound **92** after 60 min (<0.01 vs. control), and the threshold for **92** was very close to that of morphine (<0.01 vs. control) [87] (Table 2).

From *Moringa oleifera seeds*, seven new Py-2-Cs (**99**–**105**) and four known Py-2-Cs (**5**, **10**, **96**, and **106**) were isolated and characterized [88]. An oxygen deprivation experiment was carried out in the presence of **5**, **99**, and **103** at 0.1 μM using PC12 cell lines. Significant attenuation of cell death was observed for all compounds compared with the control (absence of the compound). Western blot analyses of Nrf2 and NFκB revealed that **99** and **103** upregulated Nrf2, and **5**, **99**, and **103** downregulated NFκB (Table 2). 

### 4.3. Physiological Activities of Pyrrole-2-carboxaldehydes Originating from Microorganisms

Mycalazal 2 (**122**), isolated from the sponge *Mycale micracanthoxea*, showed cytotoxic effects against the cell lines p388 (mouse lymphocyte), SCHABEL (mouse lymphocyte), A549 (human lung carcinoma), HT29 (human colon carcinoma), and MEL28 (human melanoma cell) with ED_50_ values of 2, 2, 5, 5, and 5 μg/mL respectively. Mycalazal (**121**) did not show cytotoxic effects against any of these cell lines because of its instability under the experimental conditions, and another 12 2-acyl-5-hydroxypyrrole derivatives showed moderate (in some cases strong) cytotoxic effects against these cell lines [91] (Table 3).

Thirteen new and two known Py-2-Cs (**123**–**135**, **115**, and **116**) were examined for inhibitory activity against the proliferation of *Leishmania mexicana* promastigotes. Fourteen Py-2-Cs did not show any activity (LD_50_ > 25 μg/mL), only one new compound (**123**) bearing a nitrile group at the terminus of the side chain showed moderate inhibitory activity (LD_50_ = 12 μg/mL) [92] (Table 3).

The antifungal and antiviral activity of two Py-2-Cs, 5-octadecyl Py-2-C and (6′*Z*)-5-(6′-hexeicosenyl) Py-2-C (**129**), (6′*Z*)-5-(6′-heneicosenyl)pyrrole-2-carboxaaldehyde (**137**), and other metabolites were examined for antifungal and antiviral activity. Compounds **136** and **137** did not show any significant activity. Compound **136** showed hypoglycemic activity in normal rats at an oral dose of 30 mg/kg body weight. At the same concentration, a reduction in the glucose level was also observed, which was almost equivalent to that of gilbenclamide administered orally at 30 μg/kg. Similar results were obtained in alloxan-treated diabetic rats [93] (Table 3).

Fourteen new Py-2-Cs (**141**–**154**) and eight structurally related known Py-2-Cs (**109**, **112**, **115**, **117**, **123**, **126**, **127**, and **131**) isolated from the sponge *Mycale Cecilia* collected in the Gulf of California (Mexico) were tested for cytotoxic effects against various tumor cell lines (prostate carcinoma, DU-145 and LN-caP; ovarian adenocarcinoma IGROV; breast adenocarcinoma SK-BR3; melanoma SK-MEL-28; lung adenocarcinoma A549; chronic myelogenous leukemia K-562; pancreatic carcinoma PANC-1; colon adenocarcinoma HT-29 and LOVO; colon adenocarcinoma resistant to doxorubicin LOVO-DOX; and cervix epithelial adenocarcinoma HeLa). These compounds showed moderate cytotoxic effects (the GI_50_ value was the highest for 0.2 μg/mL of **148** against LN-caP). In A-549 cells, only one compound (**142**) showed a moderate cytotoxic effect, and in LN-caP and SK-MEL cells, moderate cytotoxic effects were observed for 12 compounds. The compound **123**, which bears a nitrile functional group at the terminal carbon of the side chain, showed moderate inhibitory activity against *Leishmania mexicana;* however, compounds bearing a nitrile group on the side chain (**152**, **153**, **154**, and **123**) did not show strong activity against most of the cancer cell lines, but showed mild and selective inhibitory activity against PANC-1, LOVO, and HeLa cells. Other metabolites containing a nitrile group have shown strong biological activity [104]. The nitrile functional group is quite interesting in the pharmaceutical standpoint of view. It is well known that other metabolites bearing nitrile group in the molecule sometimes showed strong biological activities [110]. In an investigation of the cytotoxic effects of Py-2-Cs containing 21 carbon atoms, including tri-unsaturated (**141**), di-unsaturated (**144**), mono-unsaturated (**148**), and non-saturated (**109**) compounds, the mono-unsaturated Py-2-C (**148**) showed the strongest activity when compared with the other compounds (**141**, **144**, and **109**). The structure–activity relationship, including the effects in different tumor cells, of compounds **141**, **144**, and **109** has not yet been clearly established [95] (Table 3).

A lipophilic extract of the marine sponge Mycale sp. showed inhibition of the activation of HIF-1 in a T47D (human breast cancer)-based reporter assay. Bioassay-guided isolation of the extract afforded 18 new lipophilic Py-2-Cs (**155**–**172**) and eight known Py-2-Cs (**115**, **116**, **123**, **126**, **127**, **141**, **152, 136**, and **153**). The effect of these compounds on HIF-1 activation was investigated quantitatively. Two compounds (**157** and **158**) showed strong activation of HIF-1 with IC_50_ values of 7.8 (95% CI, 6.8–8.8 μM) and 8.8 μM (95% CI, 7.6–9.9 μM, respectively). Four compounds (**152**, **156**, **159**, and **164**) showed moderate activation with IC_50_ values of 10–20 μM. Seven compounds (**126**, **141**, **166**, **168**, **169**, **170**, and **172**) showed weak activation with IC_50_ values of 20–30 μM. The other 13 Py-2-Cs (**115**, **116**, **123**, **127**, **153**, **155**, **160**–**163**, **165**, **167**, and **171**) did not show any HIF-1 activation at 30 μM (<50% highest concentration in these experiments). The mechanism of action of compounds **157** and **168** was examined to determine the effect on hypoxia-induced vascular endothelial growth factor (VEGF) secretion using the T47D cell line. The Py-2-C (**150**) bearing a nitrile group at the terminus of the lipophilic side chain showed VEGF suppression at a concentration of 30 μM. The mechanism of action was investigated using T47D cells. Two compounds inhibited the T47D respiration in a concentration-dependent fashion and did not affect the mitochondrial electron transport complexes II, III, and IV. These compounds disrupted mitochondrial respiration at complex I [96]. Focusing on the diene compounds (**155**, **156**, **157**, and **158**) bearing a nitrile group at the terminal carbon, two compounds (**157**, **158**) showed strong HIF-1 activity, whereas **156** showed moderate activity. In contrast, **155** showed no activity. The number of carbons is the same in the alkyl chains of **155** and **156**; however, the number of carbons in the alkyl chain between the pyrrole ring and the C=C double bond in the side chain is 2 in **155** and 14 in **156**. For **157** and **158**, the numbers of carbons in the alkyl chain between the pyrrole ring and the C=C double bond in the side chain are 10 and 11, respectively. These differences suggested the importance of the distance between the two functional groups for the physiological activity. Similar results were also observed for compounds containing one double bond in the chain (**159**, **160**, **161**, **162**, and **163**) and bearing a nitrile group at the terminal carbon. Moderate activity was observed for **159**; however, the other compounds (**160**–**163**) did not show any activity. There are 19 carbons in the alkyl chain at the 2-position of **159**, and the other compounds (**160**–**163**) have a longer chain length than **159**. The numbers of carbons in the alkyl chain between the pyrrole ring and the C=C double bond of **159**, **160**, **161**, **162**, and **163** are 11, 13, 15, 6, and 6, respectively. This observation also suggested the importance of the length of the alkyl chain between the pyrrole ring and the C=C double bond in the side chain. Regarding the triene derivatives **171**, **172**, and **141**, two compounds (**171** and **141**) showed weak HIF-1 activity, while compound **172** with a 9-carbon alkyl chain between the pyrrole ring and the double bond in the side chain did not show HIF-1 activity. The importance of the length of the alkyl chain between the pyrrole ring and the nitrile functional group in the side chain can also be observed in the saturated alkyl nitrile compounds (**123**, **152**, **153**, **164**, and **165**). Compounds **152** and **164** showed moderate HIF-1 activity, whereas **123**, **153**, and **165** did not show any HIF-1 activity. The numbers of carbons in the alkyl chains of **164**, **152**, **165**, **123**, and **153** are 15, 17, 18, 19, and 20, respectively. The clear asynechia might be present in **152** and **165**, which resulted from the different chain lengths of these compounds (Table 3).

A 2:1 mixture of Py-2-Cs (**172** and **173**) isolated from *Mycale phyllophia* was tested for cytotoxic activity against the L5178Y cell line (mouse lymphoma cell) in vitro. The IC_50_ value (growth inhibition) of this mixture was 1.8 μg/mL [97].

Two new (**174** and **175**) and five known 5-alkyl-pyrrole carboxaldehydes (**123**, **138**, **153**, **154**, and **164**) bearing a nitrile group at the end of the carbon chain were assayed for inhibitory activity against PTP1B. Almost all the compounds showed inhibitory activity. The compounds micalenitrile-15 (**174**) and micalenitrile-16 (**175**) showed significant PTP1B131 inhibitory activity with IC_50_ values of 8.6 ± 1.9 and 10.0 ± 0.2 μM, respectively (IC_50_ = 3.6 ± 0.2 μM for the positive control ursolic acid). The known compound **138**, which has a very similar structure to micalenitrile-16 (**175**) except for the length of the side chain (which is two carbons longer), showed stronger inhibitory activity (IC_50_ = 3.1 ± 0.1 μM) compared with micalenitrile-16. This IC_50_ value was almost the same as that of the positive control (ursolic acid; IC_50_ = 3.6 ± 0.2 μM). A long alkyl side chain and the presence of a nitrile functional group might play important roles in the PTP1B inhibitory activity [99].

Thirteen alkaloids, including five pyrrol-2-carbaldehydes (**36**, **38**–**40**, and **48**), isolated from *Jishengella endophytica* 161111 were tested for antiviral effects on the H1N1 virus using the cytopathic effect inhibition assay. The cytotoxic effects of these compounds were tested in Madin Daby canine kidney (MDCK) normal cells using the MTT assay. In both assays, the five pyrrole-2-carbaldehydes (**36**, **38**–**40**, and **48**) did not show any effects (H1N1, IC_50_ > 50 μg/mL; MDCK, 50% cytotoxic concentration: not detected) [100] (Table 3).

Two Py-2-Cs (**39** and **48**) isolated from a rice-supplemented liquid medium, and other metabolites isolated from a solid or liquid medium, were tested for the ability to affect the interaction between *S. albospinus* RLe7 and the fungus *Coniochaeta* sp. Fle4, which is also an endophyte of the same plant *(Lychnophors ericoides*). Neither of these compounds showed any antifungal activity nor did they show an inducing effect on fungal pigmentation [101] (Table 3). 

Cinerol I, isolated from the sponge *Dysidea cinera*, belongs to the nitrogeneous meroterpenoid family and contains a Py-2-C moiety [98]. The inhibitory activity of cinerols, including cinerol I (**176**), toward six kinases (PTP1B, ACL, SHP1, SHP2, JAK-2, and ACCI) has been studied, and cinerol I (**176**) did not show clear inhibitory activity against any of the kinases tested. The cytotoxic effect of cinerols at a 32 μM concentration has also been investigated using human melanoma A375 and human embryonic kidney HEK293 non-tumor cell lines. However, **176** did not show cytotoxic effects against these cell lines.

Jiangrines A–F (**177**–**181** and **10**) were tested for anti-inflammatory activity by examining the ability to inhibit NO production in LPS-treated RAW 264.7 macrophage cells [105]. None of the compounds affected the cell viability at a 100 μM concentration. Jiangrines A–F showed a clear inhibitory effect on NO production. The IC_50_ values of jiangrine A (**177**), jiangrine B (**178**), jiangrines C + D (**179** + **180**), jiangrine E (**181**), and jiangrine F (pyrrolezanthine, **10**) were 97.8, 60.7, 30.1, 54.9, and 58.8 μM, respectively. The IC_50_ value of the positive control *N*-nitroso-l-arginine methyl ester (NAME) was 53.6 μM, which was similar to the IC_50_ values of jiangrine E (**181**) and jiangrine F (**10**).

The cell viabilities of three compounds (jiangrine A (**183** revised structure of **177**), jiangrine G (**182**), and **10**) were investigated in RAW 264.7 cells, and a slight inhibitory effect on the cell viability was observed for **10** at a 100 μM concentration. The addition of these compounds to LPS-treated raw cells decreased NO production along the *i*-NOS expression (dose range: 3.3–33 μM). The anti-inflammatory effects of these compounds were examined by measuring the cytokines TNF-α, IL-1β, and IL-6. The production of IL-1β and IL-6 was suppressed by the addition of **183** and **182**; however, the production of TNF-α was not observed in the LPS-treated raw 264.7 cells. The phosphorylation of p38 and p65 was inhibited by the addition of **183** and **182**. In addition, these compounds did not affect the ERK1/2 and JNK pathways, which indicated that **183** and **182** suppressed the production of IL-1β and IL-6 by blocking the p38 pathway. Compound **10,** at low (3.3 μM) to high (33 μM) concentrations, suppressed the production of TNF-α. The production of IL-1β and IL-6 was decreased only at low to medium (11 μM) concentrations of **10**. In contrast, the production of IL-1β and IL-6 increased at a high concentration (33 μM) of **10**, which suggested that **10** had a low concentration threshold [106].

## 5. Conclusions

Py-2-Cs have been isolated from many fungi, plants, and microorganisms. However, there is a clear structural difference between the pyrrole-2-carbaldehydes from microorganisms and plants (fungi). In most cases, the pyrrole-2-carbaldehydes isolated from microorganisms are *N*-unsubstituted pyrrole-2-carbaldehydes, whereas most pyrrole-2-carbaldehydes isolated from plants are *N*-substituted. Common substrates, such as **10,** have been isolated from many species. The pyrrole-2-carbaldehydes (**17**–**20**, **58**, **74**, and **77**) bearing a morpholine chromophore will attract much attention for their potential use in pharmaceutical applications. In addition, a recent report [111] concerning the use of pyrrole-2-carbaldehydes (**3**, **4**, and **8**) as biomarkers has indicated that pyrrole-2-carbaldehydes may have other future possible uses. The structure–activity relationships between the carbon chain length and the physiological activity are interesting to elucidate. For example, (as described above) Py-2-Cs bearing a nitrile group at the end of the carbon chain (**152**, **164**, **123**, **165**, **152**, and **164**) showed moderate activity (10–20 μM) in the inhibition of the activation of HIF-1 in a T47D (human breast cancer)-based reporter assay, while **123** and **165** showed very weak activity (30 μM < 50%). The numbers of carbons in the alkyl chains of **152**, **164**, **123**, and **165** are 17, 15, 19, and 18, respectively. A discontinuity is found between the carbon numbers 17 and 18. However, the reason for this discontinuity is unclear at present. Molecular calculation studies should be performed to explore this discontinuity. The isolation and identification of Py-2-Cs that show high physiological activities based on the original physiological assays are challenging, and the assay systems in most cases are not comprehensive. Thus, a particular assay system might miss active compounds, which may be found using a different assay system. In addition, most research has been performed by natural product and physiological scientists. To pursue knowledge of the structure–activity relationships, the use of computational chemistry is highly desirable. As described above, Py-2-Cs have the potential for use in various physiological fields, and further studies will provide new avenues of the use for Py-2-Cs in many academic and industrial areas, including pharmaceutical science.

## Figures and Tables

**Figure 1 molecules-28-02599-f001:**
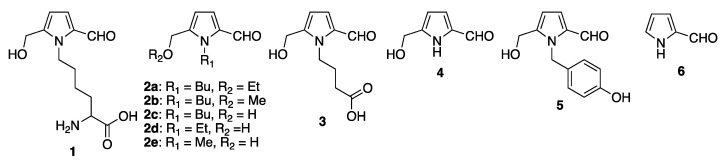
Structures of Py-2-Cs **1**–**6**.

**Figure 2 molecules-28-02599-f002:**
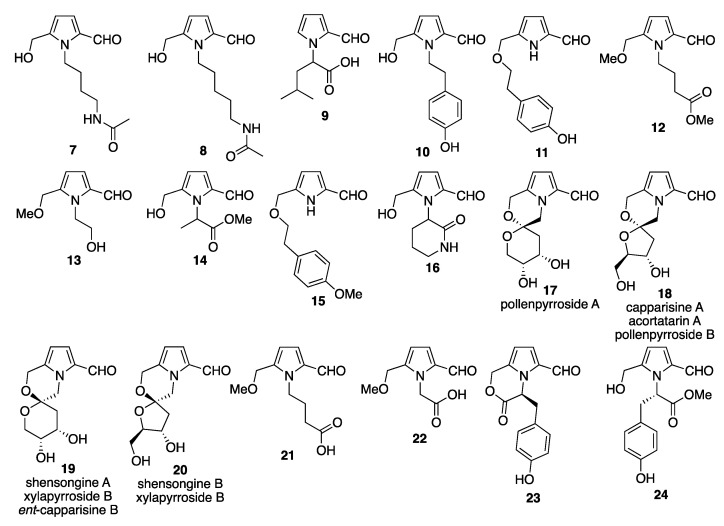
Structures of Py-2-Cs from fungi.

**Figure 3 molecules-28-02599-f003:**
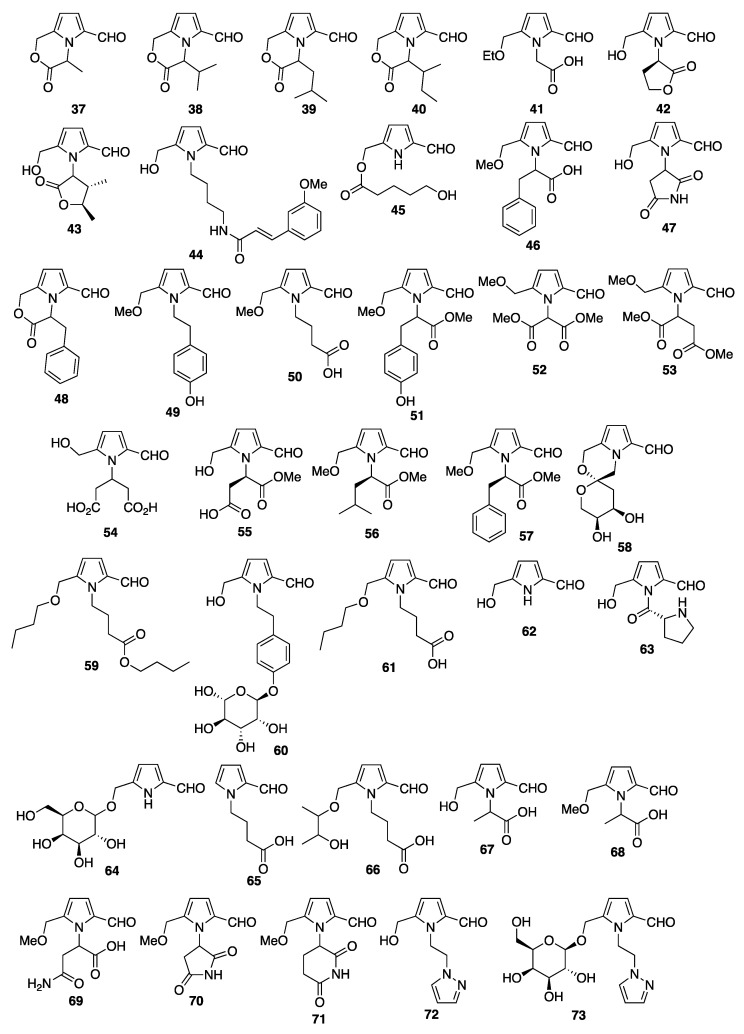
Structures of Py-2-Cs from plant sources.

**Figure 4 molecules-28-02599-f004:**
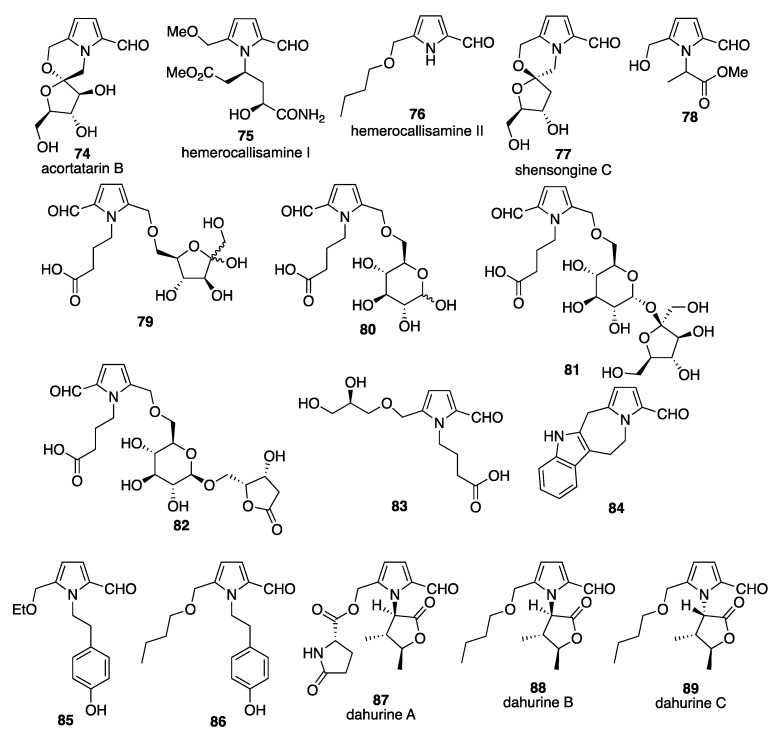
Structures of Py-2-Cs from plant sources.

**Figure 5 molecules-28-02599-f005:**
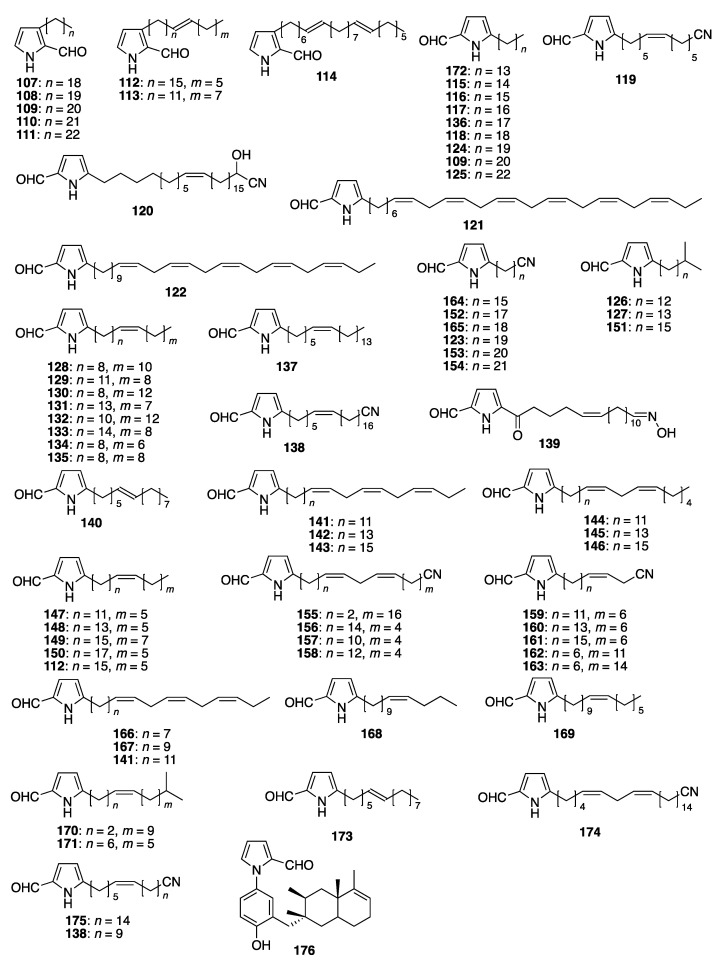
Structures of Py-2-Cs from microorganisms.

**Figure 6 molecules-28-02599-f006:**
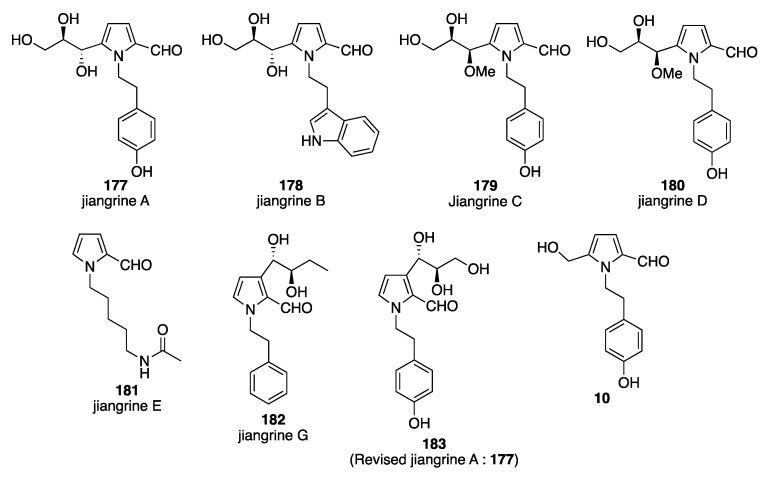
Structures of Py-2-Cs from microorganisms.

**Table 1 molecules-28-02599-t001:** Biological activity of compounds isolated from fungal species.

Fungal Species	Compound/s	Biological Activity	Strength of Activity	Reference
*Mycoleptodonoides aitchisonii*	**9**	NQO1	mild to strong	[12]
*Zizania latifolia*	**12**, **13**, **14**	osteoclast forming inhibition	very weak (no)	[18]
*Cordyceps millitaris*	**16**	adipocyte differentiation	significant	[20]
*Xylaria nigripes*	**17**, **18**, **19**, **20**	Inhibition of oxidative stress	mild to strong	[21]
*Phlebopus portentosus*	**29**, **30**, **31**, **32**, **33**, **34**, **35**	neuroprotective effect	mild to strong	[26]
*Purpureocillium lavendulum*	**37**	nematocidal activity	very strong	[107]

**Table 2 molecules-28-02599-t002:** Biological activity of compounds isolated from plant species.

Plant Species	Source	Compound/s	Biological Activity	Strength of Activity	Reference
*Magnolia coco*	leaves	**44**	inhibition off LDL oxidation	mild to strong	[108]
*Ilex paraguariensis*	leaves	**49**	elastase inhibition	no activity	[46]
*Lycium* *chinense*	fruits	**3**, **21**, **12**, **50**	hepatoprotection	very strong	[17]
*Lobelia chinensis*	MeOH extract	**61**	elastase inhibition	mild	[52]
*Aralia continentalis*	MeOH extract	**21**	elastase inhibition	mild	[54]
*Aconitum carmichaelii,*	lateral roots	**4**, **63**	antibacterial activity	weak	[56]
*Morus alba*	fruits	**3**, **4**, **21**, **65**, **66**	macrophage activation	significant (**3, 21**)	[60]
*Morus alba*	fruits	**12**, **21**, **23**, **24**, **32**, **37**–**40**, **47**, **48**, **50**, **67**–**71**	lipase activity	significant (**23**, **32**); no activity (**12**, **21**, **24**, **37**–**40**, **47**, **48**, **50**, **67**–**71**)	[109]
*Citrullus lanatus*	fruits and seeds	**72**, **73**	melanogenesis inhibition	moderate (**72**)	[61]
*Capparis spinosa*	fruits	**18**, **37**, **47**, **18**, **58**	inhibition of apoptosis	no activity	[49]
*Brassica campestris*	bee-collected	**17**, **18**	cytotoxicity toward cancer cells	no activity	[63]
*Acorus tartainowii*	rhizomes	**18**, **74**	protection against high-glucose stress (oxidative)	significant (**18**)	[66]
Daylily	flower buds	**75**, **76**	inhibition of amyloid aggregation	no activity (**75**)	[73]
*Shensong Yangxin*	capsule	**18**, **19**, **20**, **77**	cardiovascular activity	strong (**19**, **77**); no clear activity (**18**, **20**)	[75]
African mango commercial sample	powdered seeds	**3**, **4**, **37**, **78**	hydroxyl radical scavenging quinone reductase	moderate (**3**, **78**); no activity (**4**, **37**)	[76]
*Nicotiana tabacum*	leaves (EtOH)	**10**	cytotoxicity in lung cancer cells	moderate	[79]
*Juglans regia*	flowers (EtOH)	**84**, **85**	cytotoxicity (cancer cells)	significant (**84**)	[81]
*Reynoutria ciliinervis* (Nakai) moldenke	roots (EtOH)	**86**	antibacterial and antifungal	moderate	[82]
*Angelica dohurica*	Root	**59, 75**, **87**–**92**, **93**, **94**	AChE inhibition	weak (**59**, **75**, **88**, **89**, **90**)	[83]
*Selaginella delicatula*	*n*-BuOH extract	**95**	antivirus (HBV)	very weak or no activity	[85]
*Hosta Plantaginea*		**96**	anti-inflammatory	significant	[86]
*Urtica fissae herba*	aerial part	**12**, **16**, **18**, **21**, **58**, **59**, **61**, **92**, **97**, **98**	antinociception	significant (**59**, **61**, **92**, **98**)	[87]
*Moringa oleifera*	seeds	**5**, **10**, **96**, **99**–**105**, **106**	neuroprotective	significant (**5**, **99**, **103**)	[88]

**Table 3 molecules-28-02599-t003:** Biological activity of compounds isolated from microorganisms.

Microorganism	Source	Compound/s	Biological Activity	Strength of Activity	Reference
*M. micracanthoxea*	Sponge	**121**, **122**	cytotoxicity toward cancer cells	strong (**122**); no activity (**121**)	[91]
*M. microsigmatosa*	Sponge	**115**, **116**, **123**–**135**	proliferation inhibition (parasite)	moderate (**123**)	[92]
*M. mytilorum*	Sponge	**136**, **137**	hypoglycemic activity	weak (**136**)	[93]
*Mycale cecilia*	Sponge	**109**, **112**, **115**, **117**, **123**, **126**, **127**, **131**, **141**–**154**	cytotoxicity (LN-caP, GROV, SK-BR3, SK-Mel-28, A-549, K-562, PANC1, LOVO, HeLa)	strong (**148**, LN-cap); moderate (**123**, **142**)	[95]
*Micale* sp.	Sponge	**115**, **116**, **123**, **126**, **127**, **141**, **152**, **153**, **155**–**172**	HIF activity in T47D cells	strong (**157**, **158**); moderate (**152**, **156**, **159**, **164**); weak (**126**, **141**, **166**, **168**, **169**, **170**, **172**)	[96]
*M. phyllophia*	Sponge	**172**, **173** (2:1 mixture)	cytotoxicity toward Hela cells	significant	[97]
*M. lissochela*	Sponge	**123**, **138**, **153**, **154**, **164, 174, 175**	PTP1B inhibition	strong (**138**); significant (**174**, **175**)	[99]
*J. endophytica* 161111	actinomycete	**36**, **38**–**40**, **48**	anti-H1N1, cytotoxicity (kidney cells)	no activity	[100]
*S. albospinus* RLe7	Endocyte	**39**, **48**	antifungal, fungal pigmentation inhibition	no activity	[101]
*D. cinera*	Sponge	**176**	PTP1B inhibition	no activity	[98]
*Jiangella gansuensis*	actinobacterium	**10**, **177**–**181**	inhibition of macrophage NO production	strong (**10**, **177**–**181**)	[105]
*Jiangella alba*	fermentation broth	**10**, **182**, **183**	anti-inflammatory	Strong (**10**, **182**, **183** *)	[106]

* Mechanism is different.

## Data Availability

Not applicable.

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
