# Peer review of "Pyrrole-2-carboxaldehydes: Origins and Physiological Activities"

_molecules, 2023, doi:10.3390/molecules28062599_

Round 1

Reviewer 1 Report

This is an interesting and novel review article which should be of great interest to the readership of Molecules.

The physiological role of pyrrole-2-carboxaldehydes is a field of increasingly recognized importance.

I have the following suggestions to improve the quality of this paper:

1. Extensive editing of the English language and grammar is needed (compulsory). Some sentences are difficult to understand.

I would like to suggest that the author consult a native speaker or edit the manuscript via an eidting service.

2. In addition to the many chemical structures authors need to provide a couple of figures (e.g. 4-5) to summarize the various physiological properties of pyrrole-2-carboxaldehydes  (compulsory).

3. A chapter regarding future directions of research is missing but would improve the quality of this paper (compulsory).

Author Response

1, English revision

              According to the reviewer’s suggestion, we revised the English by the help of the English proofreading company (native English speaker). According their suggestions and comments, we revised the manuscript in almost rewritten level.

2, Figure and Table revision

              According to the reviewer’s suggestion, we made Fig. 5 compact, further, for the better understanding of the readers, we added the reference number in Tables (Tables 1-3).

3, Future direction of the research

     According to the reviewer’s suggestions, we revised the conclusion part as shown below.

Py-2-Cs have been isolated from many fungi, plants, and microorganisms. However, there is a clear structural difference between the pyrrole-2-carbaldehydes from microorganisms and plants (fungi). In most cases, the pyrrole-2-carbaldehydes isolated from microorganisms are N-unsubstituted pyrrole-2-carbaldehydes, while most pyrrole-2-carbaldehydes isolated from plants are N-substituted. Common substrates, such as 10, have been isolated from many species. The pyrrole-2-carbaldehydes (17-20, 58, 74, and 77) bearing a morpholine chromophore will attract much attention for their potential use in pharmaceutical applications. In addition, a recent report [112] concerning the use of pyrrole-2-carbaldehydes (3, 4, and 8) as biomarkers has indicated that pyrrole-2-carbaldehydes may have other future possible uses. The structure–activity relationships between the carbon chain length and the physiological activity are interesting to elucidate. For example, (as described above) Py-2-Cs bearing a nitrile group at the end of the carbon chain (152, 164, 123, 165, 152, and 164) showed moderate activity (10–20 mM) in the inhibition of the activation of HIF-1 in a T47D (human breast cancer)-based reporter assay, while 123 and 165 showed very weak activity (30 mM < 50%). The numbers of carbons in the alkyl chains of 152, 164, 123, and 165 are 17, 15, 19, and 18, respectively. A discontinuity is found between the carbon numbers 17 and 18. However, the reason for this discontinuity is unclear at present. Molecular calculation studies should be performed to explore this discontinuity. The isolation and identification of Py-2-Cs that show high physiological activities based on the original physiological assays are challenging, and the assay systems in most cases are not comprehensive. Thus, a particular assay system might miss active compounds, which may be found using a different assay system. In addition, most research has been performed by natural product and physiological scientists. To pursue knowledge of the structure–activity relationships, the use of computational chemistry is highly desirable. As described above, Py-2-Cs have the potential for use in various physiological fields, and further studies will provide new avenues of the use of Py-2-Cs in many academic and industrial areas, including pharmaceutical science.

By changing the conclusion part, we believe that this manuscript will stimulate the willingness of the researchers working in the medicinal, natural product, and physiological scientific areas.

Author Response

Thank you for your comments and opinions of our manuscript. We rewrite the manuscript significantly including English revision.

1, the word pyrrole-2-carboxaldehide is repeated a lot of times in all text (May be you can use instead Py-2-Cor something like this)

According to your suggestion, the word “pyrrole-2-carboxaldehyde” was abbreviated as Py-2-C, except in the title (subtitle).

2, -Is necesary the word chromophore in pyrrole-2-carboxaldehide chromophore?

According to your suggestion, we deleted the word “ chromophore “.

3, -The compound numbers should be reffered to the respective figure in the text.

Thank you for your comments, for the better understanding of the readers, we added the reference number of the compounds in the Table. We also summarize the figures based on the chemical structure of the molecule.

4, Line 33. What is crossline in the sentence?

Thank you for your comment, we deleted the crossline.

5, Line 49. In the case of yields of compounds, indicate how much.

In figure 1. Nine compounds are produced? They should be 2a-h.

Thank you for your valuable comments, we added the yield in the text. We also revised the Figure 1 based on the literature.

6, Line 54. Hydroxymethyl derivatives as 3-5?.

Thank you for your valuable suggestions, TLC-analysis based formation of hydroxylmethyl derivatives were proposed but not isolated. In most cases, the authors used 2,4-dinitrophenylhydrazine (DNP) to stabilize the reaction products, however, the use of DNP caused further reactions, such as dehydration.

7, Line 121. Along with leccinitine A, should be eliminated.

Thank you for your comment, we eliminated “Along with leccinitine A” in the text.

8, Line 129. Include references in the case of compounds previously reported (all the text)

Thank you for your comments, according to your suggestion, we added the references in the text. (one compound had already been chemically synthesized [5]) .

9, Line 151-153. There are two or six unknown compounds?.

Thank you for your comments, we revised the manuscript as follows.

--two unknown (23, and 24) and six known Py-2-Cs were isolated, along with flavonoid derivatives. The chemical structures of the known Py-2-Cs (25, 26, 27, 3, 21, and 12) were determined based on ---.

10, Line 165. Isolated instead of obtained in all the text.

Thank you for your comment, according to your suggestion, we changed the word “obtain” to isolate.

11, Line 171. Compound (36) instead of compound (37)

Thank you for your comment, we change the compound number 36.

12, Line 178. What is chromore?.

Thank you for your comment. We changed the “lactone structure” instead of lactone chromophore.

13, Line 192. Various spectroscopies instead of various spectroscopy.

Thank you for your comments, we changed “---characterized by various spectroscopic methods” instead of various spectroscopy.

14, Line 209-211. Compoun 3 do not have methyl and methoxyl groups. Compound 46 do not have methyl group. Check the names of all compounds in the text and homogeinize the nomenclature.

Thank you for your comments, we revised the name of the compound. According to your suggestions, we revised the sentences as follows. ----as 4-(2-formyl-5-methoxymethylpyrrol-1-yl) butyric acid (3) and 2-(2-formyl-5-methoxymethylpyrrol-1-yl)-3-phenylpropionic acid (46).

In the text, we basically use the names based on the author’s nomenclature except the wrong use of author’s nomenclature.

15, Line 220. 5-(methoxymethyl)1H-pyrazole-2-carboxaldehide not corresponds to compound 37.

Thank you for your comments, we revised the sentences as follows.

In 2005, from the ethanol extract of Salvia miltiorrhiza, five nitrogen-containing compounds, including 5-(methoxymethyl)-1H-pyrrole-2-carbaldehyde (36), have been isolated and characterized spectroscopically [43]

16, Line 320. Compound 37 is wrong named.

Thank you for your comments, we revised the manuscript as follows.

From the fruit of Capparis pinosa, two new Py-2-C derivatives—capparisine A (18) and capparisine B (58)—were isolated, along with the known Py-2-C derivatives 2-(5-hydroxymethyl-2-formyl-1-yl) -methyl-acetic acid lactone (37) and N-(3’-maleimidyl)-5-hydroxymethyl-2-pyrrole formaldehyde] (47) [66].

17, Line 718. Compound 84 is not a tetrahydropyrrolo.

Thank you for your comments,

We re-examined the literature, the authors used this nomenclature in their paper. The nomenclature is based on IUPAC nomenclature.

18, In the tables, the references should be included.

Thank you for your comment, as written in the previous section, we added reference number of each compound.

19, The abstract should be clearly re-structured highlighting the biological importance of pyrrole-2-carboxaldehides and including examples as pyrraline considering the importance of their biological activities.
Thank you for your valuable comments, according to your comments, we re-write the abstract as follows.

Pyrrole-2-carboxaldehyde (Py-2-C) derivatives have been isolated from many natural sources, including fungi, plants (root, leaf, and seed), and microorganisms. The well-known diabetes molecular marker, pyrraline, which is produced after sequential reactions in vivo, has a Py-2-C skeleton. Py-2-Cs can be chemically produced by the strong acid-catalyzed condensation of glucose and amino acid derivatives in vitro. These observations indicate the importance of the Py-2-C skeleton in vivo and suggest that molecules containing this skeleton have various biological functions. In this review, we have summarized Py-2-C derivatives based on the origin. We also discuss the structural characteristics, natural sources, and physiological activities of isolated compounds containing the Py-2-C group

20, In this review, synthetic methodologies as Amadori and Millard reactions are considered, including some synthetic mechanistic transformations, however, no schemes are included.

Thank you for your valuable comments, it is quite important to write the synthetic mechanistic transformations of Amadori and Millard reactions. The reaction takes much time and procedures, there are so many compounds produced in the reaction course. In this meaning, it is difficult to write a single mechanism for the formation of each compound. In this review article, we basically focus our attention on the natural products, in this meaning, we briefly describe something about the Amadori and Millard reaction in the text.

We also recognize the importance to clarify the Amadori and Millard reaction mechanisms, at present, it is out of our power.

Round 2

Reviewer 2 Report

The authors made the corrections